# Neoplastic signatures: Comparative proteomics of canine hepatobiliary neuroendocrine tumors to normal niche tissue

Tifini L. Batts[1¤a], Emi Sasaki[2], Mayzie Miller[1¤b], Joshua Sparago[1¤c], Rudy W. Bauer[2], Daniel Paulsen[2], Bonnie Boudreaux[1], Chin-Chi Liu[1], Stephanie D. Byrum[3,4], Andrea N. Johnston[1] *

1 Veterinary Clinical Sciences Department, Louisiana State University School of Veterinary Medicine, Baton Rouge, Louisiana, United States of America, 2 Department of Pathobiological Sciences and Louisiana Animal Disease Diagnostic Laboratory, Baton Rouge, Louisiana, United States of America, 3 Department of Biochemistry and Molecular Biology, University of Arkansas for Medical Sciences, Little Rock, AR, United States of America, 4 Arkansas Children's Research Institute, Little Rock, AR, United States of America

¤a Current address: MedVet, Houston, TX, United States of America
¤b Current address: Auburn University College of Veterinary Medicine, Auburn, AL, United States of America
¤c Current address: MedVet, Mountain View, CA, United States of America
* johnston1@lsu.edu

**Data Availability Statement:** All relevant data are within the paper and its Supporting Information files. Data are available via ProteomeXchange with identifier PXD028822.

## Abstract

Hepatobiliary neuroendocrine neoplasms are rare cancers in humans and dogs. To date, no large-scale primary hepatobiliary neoplasm omics analyses exist in any species. This limits the development of diagnostic biomarkers and targeted therapeutics. Neuroendocrine cancers are a heterogenous group of neoplasms categorized by their tissue-of-origin. Because the anatomic niche of neuroendocrine neoplasms shapes tumor phenotype, we sought to compare the proteomes of 3 canine hepatobiliary neoplasms to normal hepatobiliary tissue and adrenal glands with the objective of identifying unique protein signatures. Protein was extracted from formalin-fixed paraffin-embedded samples and submitted for tandem mass spectroscopy. Thirty-two upregulated and 126 downregulated differentially expressed proteins were identified. Remarkably, 6 (19%) of the upregulated proteins are correlated to non-hepatobiliary neuroendocrine neoplasia and 16 (50%) are functionally annotated within the exosome cellular compartment key to neuroendocrine signaling. Twenty-six (21%) downregulated proteins are enriched in metabolic pathways consistent with alterations in cancer. These results suggests that characteristic neoplastic protein signatures can be gleaned from small data sets using a comparative proteomics approach.

## Introduction

Neuroendocrine neoplasms (NEN), originally referred to as carcinoids, occur uncommonly in humans and dogs. Data from the National Cancer Institute's Surveillance, Epidemiology, and

**Funding:** AJ - NIH grant 1P20GM135000-01A1 and the Louisiana State University, Department of Veterinary Clinical Sciences CORP grant SB - Proteomic analysis was funded by the National Institute of General Medical Sciences (NIGMS) grant R24GM137786. SB- Arkansas Children's Research Institute, the Arkansas Biosciences Institute, and the Center for Translational Pediatric Research funded under the National Institutes of Health National Institute of General Medical Sciences (NIH/NIGMS) grant P20GM121293 and the National Science Foundation Award No. OIA-1946391.

**Competing interests:** The authors have declared that no competing interests exist.

Results Program (SEER 18) found that the neuroendocrine tumor incidence was 6.98 per 100,000 cancer patients and that the prevalence of primary hepatobiliary NENs was less than 1% of all neuroendocrine cancers [1–3]. Histologically, differentiated neuroendocrine cells are characterized by cytoplasmic neurosecretory granules, such as chromogranins or synaptophysin [4]. The International Agency for Research on Cancer—World Health Organization (IARC-WHO) has recently reclassified NENs to encompass both well-differentiated neuroendocrine tumors (NETs) and poorly differentiated neuroendocrine carcinomas (NECs) [5–7]. NETs and NECs are identified morphologically and graded using replicative indices (mitotic count, Ki-67%) [8]. NETs are graded G1 to G3 and NECs are uniformly G3. Despite these guidelines, global characterization of these tumors remains challenging. This is partly because NENs are a heterogeneous group of tumors that arise from anatomically diverse regions [5,9,10]. Neuroendocrine neoplasms maintain genotypic and phenotypic characteristics of these niches. Yet, certain genetic mutations, such as *MEN1*, occur across NEN subtypes and can correlate with survival outcomes [11,12]. Proteogenomic profiles will be central to NEN characterization and will ultimately define therapeutic targets.

In dogs, reports of hepatobiliary NENs appear sporadically in the literature [4,13–17]. Two descriptive case series encompassing 23 cases have illustrated the clinical features, immunohistochemical markers, and ultrastructural traits of canine gallbladder and liver NENs [4,13]. Despite these summaries, the rarity of hepatobiliary NENs in dogs limits a complete characterization these tumors and their clinical progression. Reported outcomes are variable, but dogs with gallbladder NEN have longer survival times than dogs with hepatic NEN. Additional diagnostic biomarkers are needed to predict survival and generate informed treatment protocols. To accomplish this objective, we utilized formalin-fixed paraffin embedded samples of 3 canine hepatobiliary neuroendocrine tumors to define the differential protein expression compared to normal liver and adrenal glands from unrelated, adult dogs. The rationale for comparison of hepatobiliary NENs to normal tissue was that neoplastic transformation would promote differential protein expression distinct from the hepatobiliary tissue of origin and non-transformed neuroendocrine tissue.

## Materials and methods

### Ethics statement

We did not seek prospective approval because this study did not meet the requirements for IACUC at our institution. This study was retroactively exempted by IACUC Committee with Animal Welfare Assurance #A3612-01, License #72–3, and Multiple Assurance #M1128.

### Patients and samples

Formalin-fixed, paraffin-embedded (FFPE) tissues from 2 canine gallbladder NENs and 1 primary hepatic NEN were collected. The patients with gallbladder NENs were a 5-year-old, male neutered Boston terrier and a 7-year-old, male neutered Doberman pinscher who underwent cholecystectomies for tumor removal. Liver biopsies were obtained from both patients. The Boston terrier was asymptomatic with a normal complete blood count and serum biochemistry. The Doberman pinscher presented for hematemesis after dietary indiscretion and had a mild elevation in GGT and cholesterol with moderate to marked elevations in ALP and ALT, respectively. No extrahepatobiliary masses were identified on abdominal ultrasound or during surgery. The patient with the liver NEN was a 12-year-old, male neutered Rat terrier with a history of lethargy, polyuria, polydipsia, and diarrhea. He had a normal complete blood count and moderate elevations of ALP and ALT on serum biochemistry. The tissue from this tumor was obtained by surgical biopsy. The diffuse nature of the mass precluded complete resection;

however, gross extrahepatic metastases were not identified on abdominal CT nor at exploratory laparotomy. Liver tissue and adrenal glands from three normal adult canines were obtained from the Louisiana Animal Disease Diagnostic Laboratory histology repository. Hematoxylin and eosin (H&E) stains were performed on all tissues. Sections were reviewed by 3 anatomic pathologists (ES, DP, RB).

## Immunohistochemistry and immunofluorescence

The NENs were additionally immunohistochemically stained with chromogranin A (Dako A0430), synaptophysin (Abcam ab8049), neuron specific enolase (Dako M0873), Cytokeratin 7, and Ki-67. Four-micron thick sections of FFPE tissues were mounted on positively charged Superfrost® Plus slides (VWR, Radnor PA) and subjected to IHC using the automated BOND-MAX and the Polymer Refine Detection kit (Leica Biosystems, Buffalo Grove, IL). Following automated deparaffinization, heat-induced epitope retrieval (HIER) was performed using a ready-to-use citrate-based buffer (pH 6.0; Leica Biosystems) at 100°C for 20 minutes before incubation for all antibodies except synaptophysin. Sections were incubated with the primary antibodies for 30 minutes at room temperature, followed by either a polymer-labeled goat anti-rabbit or anti-mouse IgG coupled with horseradish peroxidase (Leica Biosystems) for 20 min at room temperature. Three, 3'-diaminobenzidine tetrahydrochloride (DAB) was used as the chromogen (10 minutes), and counterstaining was performed with hematoxylin. Slides were mounted with a permanent mounting medium (Micromount®, Leica Biosystems). Gastrin immunostaining was performed on the gallbladder NEN by the Animal Health Diagnostic Center at Cornell University College of Veterinary Medicine. Immunofluorescent labeling was performed using anti-galectin-1 antibody (Genetex 89349–128) and anti-transgelin (Genetex 113561). Five μm sections were mounted on charged slides. Samples were deparaffinized in xylene and serially rehydrated using a descending gradient of ethanol-water solutions. Slides were washed in phosphate buffered saline with 0.1 % Triton X-100. After citrate buffered antigen retrieval, tissues were blocked with 5 % normal goat serum at room temperature for one hour. Tissues were incubated with primary antibody in 1 % in normal goat serum at room temperature for 2 hours, washed, and incubated with secondary antibody (Biotium CF594 F(ab') or Invitrogen A3271 Alexa Fluor Plus 488) for 1 hour at room temperature. Hoechst 33,342 (H42) nuclear stain (1ug/mL, Thermo Scientific) was applied for 20 minutes at room temperature. After dye incubation, slides were washed for 5 minutes with PBS. Slides were cover slipped using the Biotium EverBrite Hardset Mounting Medium (Fremont, CA) and allowed to dry for a minimum of 30 minutes prior to microscopic analysis. All slides were reviewed by 3 veterinary anatomic pathologists (ES, DP, RB). Ki-67 proliferation index value was determined by counting 500 cells in defined regional hot-spots.

## Quantitative proteomics

A 5 micron scroll of each FFPE tissue was submitted to The IDeA National Resource for Quantitative Proteomics (http://idearesourceproteomics.org/) for processing. Briefly, FFPE tissue scrolls were deparaffinized by solubilization in xylene. Extracted proteins were then reduced, alkylated, and digested using filter-aided sample preparation with sequencing grade modified porcine trypsin (Promega) [18]. Tryptic peptides were labeled using tandem mass tag isobaric labeling reagents (Thermo) following the manufacturer's instructions and combined into one 10-plex sample group. The labeled peptide multiplex was separated into 46 fractions on a 100 x 1.0 mm Acquity BEH C18 column (Waters) using an UltiMate 3000 UHPLC system (Thermo) with a 50 min gradient from 99:1 to 60:40 buffer A:B (Buffer A is 0.1% formic acid, 0.5% acetonitrile and Buffer B is 0.1% formic acid, 99.9% acetonitrile) ratio under basic

pH (pH 10) conditions, and then consolidated into 18 super-fractions. Each super-fraction was then further separated by reverse phase XSelect CSH C18 2.5 um resin (Waters) on an in-line 150 x 0.075 mm column using an UltiMate 3000 RSLCnano system (Thermo). Peptides were eluted using a 60 min gradient from 98:2 to 60:40 buffer A:B ratio. Eluted peptides were ionized by electrospray (2.2 kV) followed by mass spectrometric analysis on an Orbitrap Eclipse Tribrid mass spectrometer (Thermo) using multi-notch MS3 parameters. MS data were acquired using the FTMS analyzer in top-speed profile mode at a resolution of 120,000 over a range of 375 to 1500 m/z. Following CID activation with normalized collision energy of 35.0, MS/MS data were acquired using the ion trap analyzer in centroid mode and normal mass range. Using synchronous precursor selection, up to 10 MS/MS precursors were selected for HCD activation with normalized collision energy of 65.0, followed by acquisition of MS3 reporter ion data using the FTMS analyzer in profile mode at a resolution of 50,000 over a range of 100–500 m/z. The mass spectrometry proteomics data have been deposited to the ProteomeXchange Consortium via the PRIDE partner repository with the dataset identifier PXD028822 and 10.6019/PXD028822 [19].

## Data analysis

Tandem mass spectrometric data were searched with MaxQuant (Max Planc Institute; version 1.6.17.0) against the UniprotKB/SwissProt Canis familiaris protein database (March 2020) for protein identification. Proteins were normalized using normalization in the ProteiNorm app [20]. Proteins with significant abundance differences when comparing NEN versus normal adrenal and NEN versus normal liver tissues were identified by performing an empirical Bayes moderated t-test using the BioConductor package Limma v 3.36.2 and Benjamini–Hochberg FDR correction [7]. Unless otherwise stated, an FDR cutoff of 0.05 was applied. To identify differentially abundant tumor proteins compared to adrenal glands and liver tissue, we filtered protein expression by fold change ($>$ 2). Protein or peptide clustering was performed with Cluster 3.0 (http://bonsai.hgc.jp/~mdehoon/software/cluster/software.htm) using hierarchical clustering and Euclidian distance as the metric [8]. Functional enrichments of gene ontology (GO) categories from *Canis familiaris* were performed using ShinyGO v.06 using a p-value cutoff of 0.05 (http://bioinformatics.sdstate.edu/go/) [14,21]. The Database for Annotation, Visualization, and Integrated Discovery v6.8 (DAVID, https://david.ncifcrf.gov/) was used to identify the enriched biological processes, cellular component, molecular function, and KEGG pathways [22,23]. Protein-protein interaction (PPI) networks were generated using the online Search Tool for the Retrieval of Interacting Genes (STRING; https://string-db.org/) and Cytoscape StringApp (http://apps.cytoscape.org/apps/stringapp) [24]. Differentially expressed upregulated proteins were evaluated for expression in neuroendocrine tumors including carcinoids using the Human Protein Atlas (https://www.proteinatlas.org/) [25].

## Results

### Histology and immunohistochemistry

The 3 neuroendocrine neoplasms had a similar histologic appearance on H&E staining; neoplastic polygonal cells forming dense sheets, nests, and packets, on a fibrovascular stroma (Figs 1 and 2A). Both gallbladder NENs had evidence of vascular invasion on gallbladder histology, but metastatic neoplasia was not identified in the liver on histology nor grossly in other organs. Consistent with the diagnosis of neuroendocrine neoplasia, all NENs displayed immunoreactivity to chromogranin A, neuron specific enolase, and synaptophysin (Fig 2B–2D). Additionally, gallbladder NENs were not immunoreactive to the biliary epithelial marker cytokeratin 7 and, in contrast to previous studies, gallbladder NENs were not immunoreactive to gastrin

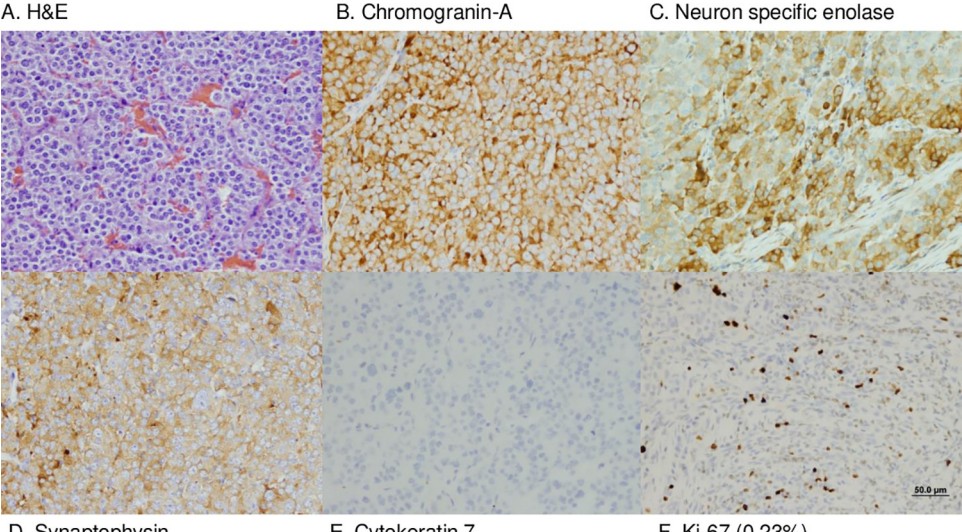

**Fig 1. Neuroendocrine neoplasm sections submitted for quantitative proteomics.** H&E stained histologic sections of the 2 gallbladder neuroendocrine neoplasms (left panel, middle panel) and the hepatic neuroendocrine neoplasm (right panel). Scrolls of these sections were submitted for proteomic analysis.

(Fig 2E, not shown) [13]. In all NENs, the Ki67 index was less than 3% (gallbladder: 0.04%, 0.05%; liver: 0.23%; Fig 2F). Based on the human WHO classification system, all neuroendocrine neoplasms in this study met the criteria for NETs G1. Survival time of the dogs with gallbladder NENs exceeds 730 days. After cholecystectomy, the Doberman Pinscher underwent no further therapy, and the Boston terrier was treated with 5 doses of carboplatin. The Rat terrier with hepatic NEN died acutely 3 days after surgery. The cause of death was undetermined but was suspected to be a vascular event. Although tumor grade has been correlated to outcome in humans, in this small canine cohort meaningful correlation of Ki67 index to survival provides limited information.

## Proteomic profile in NENs versus normal tissue

In total, 4,533 proteins were quantified. We identified 158 differentially expressed proteins, comprised of 32 upregulated proteins and 126 downregulated differentially expressed proteins (Fig 3). Hierarchical clustering of significantly expressed proteins in the three NENs identified

A. H&E  B. Chromogranin-A  C. Neuron specific enolase

D. Synaptophysin  E. Cytokeratin 7  F. Ki-67 (0.23%)

**Fig 2. Neuroendocrine neoplasm H&E histology and immunohistochemical staining.** H&E stained gallbladder neuroendocrine neoplasm (A). Neoplastic cells exhibited positive immunoreactivity for chromogranin A (B), neuron specific enolase (C), and synaptophysin (D); cytokeratin 7 staining was negative (E) and few neoplastic cells were Ki-67 positive (F). Scale Bar = 50μm.

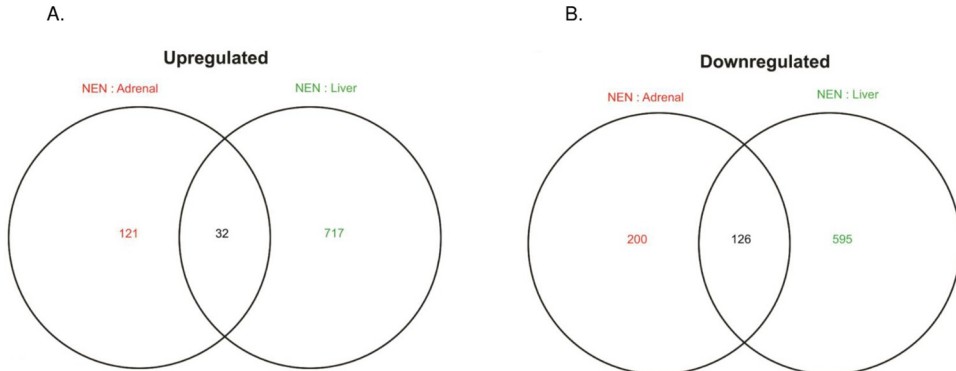

**Fig 3. Venn diagram of overlapping upregulated and downregulated proteins.** There were 32 upregulated (A) and 126 downregulated (B), overlapping differentially expressed proteins. when the NEN proteomes were compared to normal adrenal (red text) and liver tissue (green text). Up/downregulated proteins were statistically significant if fold change > 2, p-value < 0.05.

a similar proteomic profile (Fig 4A). Principle component analysis (PCA) of the NEN, normal liver and adrenal tissue proteomes demonstrates tissue specific clustering (Fig 4B).

## Protein-protein interaction networks and pathway analyses of differentially expressed proteins

STRING and the Cytoscape StringApp were employed to construct relationships among differentially expressed proteins. Among the upregulated proteins, only 3 edges were identified linking the upregulated hub proteins LYN, GNAI2, GNG10 (high confidence cutoff 0.7, Fig 5). The Src-family tyrosine kinase, LYN, is a proto oncogene that interacts with proto oncogene inhibitory guanine nucleotide (G) binding protein alpha 2 (GNAI2) involved in the hormonal regulation of adenylate cyclase and G protein subunit gamma 10 required for GTPase activity

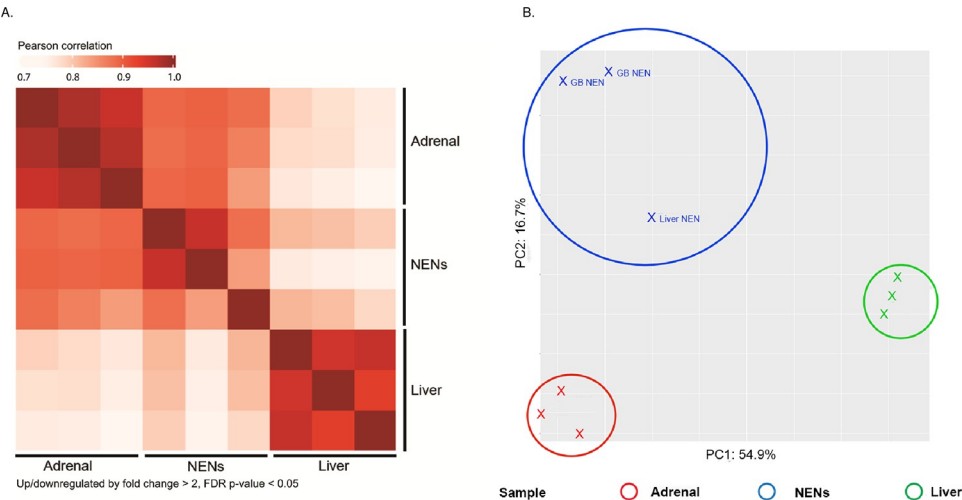

**Fig 4. Heat map of protein expression and principal component analysis.** Heat map of protein abundance patterns in differentially expressed proteins. Normal adrenal tissue is in the first row and column. Normal liver tissue is in the third row and column. The neuroendocrine neoplasms (NENs) are in the middle row and column. The darker red color indicates protein expression correlation (A). Up/downregulated by fold change >2 & FDR p-value <0.05. Principal component analysis (PCA) of normalized gene counts segregated by tissue (B).

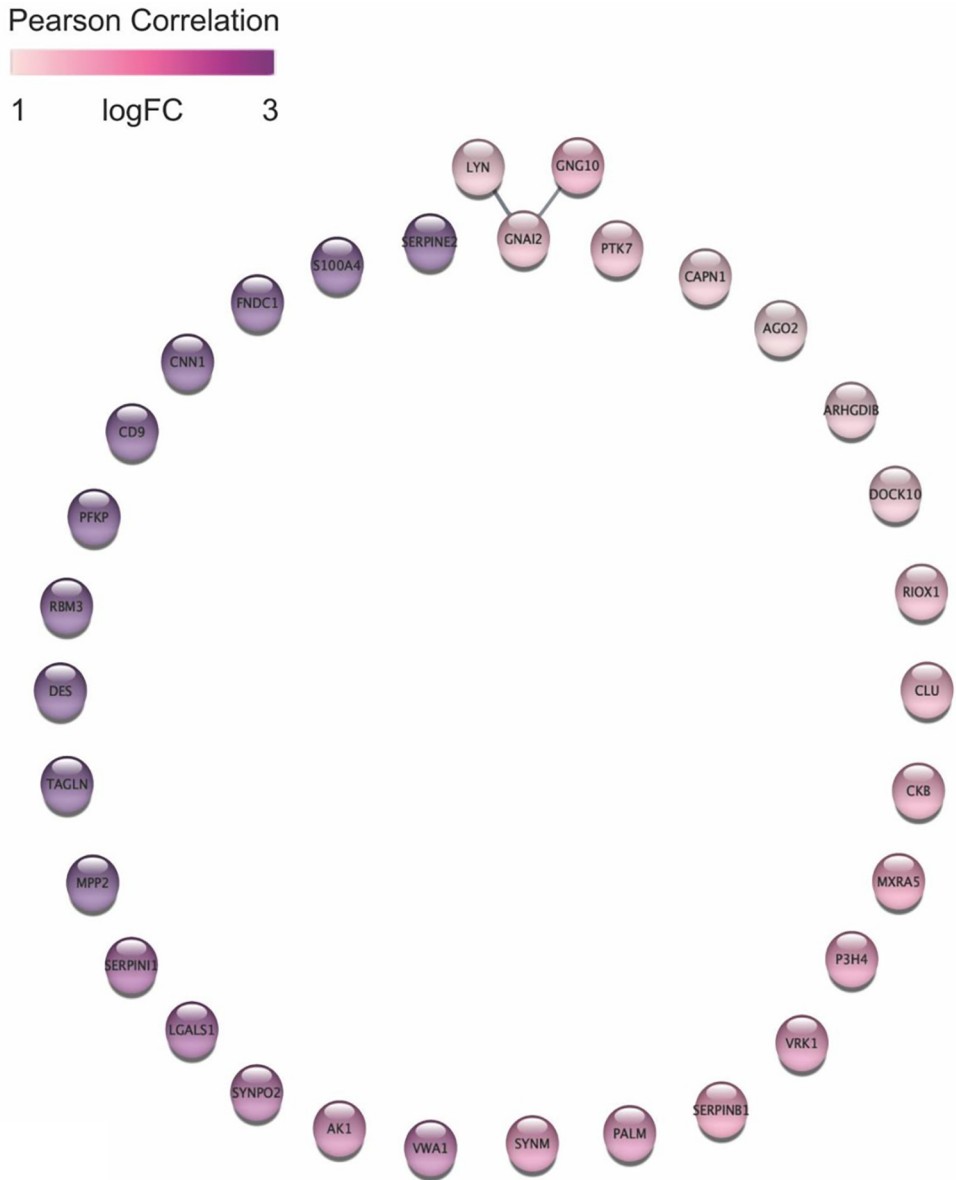

**Fig 5. All differentially expressed upregulated proteins with STRING network in canine hepatobiliary NEN.** Log-ratios of protein abundance between cancer and normal tissues were mapped using a white-purple gradient. STRING network confidence cutoff is 0.7. Image generated in Cytoscape StringApp.

and amplified in melanoma (GNG10) [26–29]. The small number of upregulated proteins limited enrichment of significantly associated biological pathways in ShinyGO using the *Canis familiaris* data set, but when all available gene sets were evaluated, the upregulated proteins clustered with hemostasis, protein complex binding, neurogenic signaling pathways, and trophoblast migration. In DAVID gene ontology cellular component analysis, 16 upregulated proteins (VWA1, PFKP, DES, DOCK10, CD9, CAPN1, GNAI2, ARHGDIB, S100A4, MXRA5, AK1, GNG10, SERPINB1, LYN, SERPINI1, CLU) were localized to exosomes, a critical neuroendocrine signaling mechanism (GO:0070062, P-value 6.5E-7). A search of The Human Protein Atlas, identified 20 upregulated differentially expressed proteins associated with cancer

(AK1, CLU, CKB, CNN1, DES, FNDC1, GNAI2, GNG10, LGALS, LYN, P3H4, PTK7, RPM3, S100A4, SYNM, SYNPO2, TAGLN, SERPIN1, SERPINB1, VWA1) and 6 that are specifically upregulated in non-hepatobiliary NENs (CKB, CLU, LGALS1, LYN, SERPINB1, TAGLN) [25,30–36]. Of these, transgelin (TAGLN) and galectin-1 (LGALS1) protein expression were validated in canine NENs by immunohistochemistry (S1 and S2 Figs).

Fifty-six edges were identified connecting downregulated protein hubs and statistical analysis suggested a biological connection (average node degree: 1.15, average local clustering coefficient: 0.247, PPI enrichment P-value <1.0E-16, Fig 6). Using ShinyGO, hierarchical clustering of the functional GO biological processes of differentially expressed downregulated proteins identified involvement in mitochondrial metabolic pathways (P-value cutoff 0.05, *Canis familiaris*). Using DAVID, 26 downregulated (DHCR24, HMGCL, ALG2, ATP6, ATPf, COASY, AK2, AK4, ALDH4A1, COX4I1, DBT, ECHS1, FH, HADHA, HADHB, HSD17B6, KYAT3, LPIN3, MMAB, MECR, MAOB, NNT, SDSL, SDHB, SUCLG1, XDH) proteins were functionally annotated to metabolic pathway and 25 were localized to the mitochondrion (DECR1, ATP6, ABHD11, ACAD9, ACSF2, DIABLO, ECHS1, FDX1L, FH, HADHB, HSDL2, KYAT3, LACTB, LDHD, MMAB, MECR, PDF, SCO1 homolog, RMND1, SDSL, SUOX, TXN2, TIMMDC1, TMEM126A, ZADH2) (KEGG_PATHWAY, P-value 2.8E-10; GO: 0005739, P-value 2.3E-12). Eight downregulated hub proteins (ACAA2, DECR1, ECHS1, HADH, HADHA, HSDL2, J9NZC6 (uncharacterized thiolase protein family member), ACSF2, MECR) were enriched within the Mitochondrial Fatty Acid Beta-Oxidation reactome pathway (STRING, CFA-8978868, strength 1.75, false discovery rate 6.95E-08).

## Discussion

Hepatobiliary NENs are uncommonly diagnosed in dogs [4,13]. As advanced imaging modalities are more widely used in veterinary patients, the relative incidence of neuroendocrine neoplasms is likely to increase as it has in humans [1]. The biological behavior of canine hepatobiliary NENs is incompletely characterized. Variable survival outcomes have been reported in previous case series may be dependent on morphology, mitotic rate, stage, hormone secretory profile, tumor niche or yet undetermined factors [13]. Application of a standardized classification system based on histology, replicative indices, and molecular expression profiles will standardize phenotyping of canine neuroendocrine neoplasms and enable correlation to patient survival rates and target therapy. To better characterize these rare tumors, we sought to compare differential protein expression between hepatobiliary neoplasms and tissue from normal canine liver and adrenal glands, with the latter representative of a heterogeneous population of neuroendocrine cell types. Few primary neuroendocrine cells exist in the hepatobiliary system [16]. NEN may result from neuroendocrine differentiation due to chronic inflammatory disease or neoplasia, but the molecular mechanisms remain unclear. Because NENs reportedly maintain protein expression patterns akin to their tissue of origin, both a neuroendocrine organ, the adrenal gland, and the liver were selected as control tissues.

The proteomic strategy resulted in identification 32 upregulated and 126 downregulated differentially expressed proteins in the neuroendocrine neoplasms. Although the upregulated protein pool was relatively small, protein species including AK1, CLU, CKB, CNN1, DES, FNDC1, GNAI2, GNG10, LGALS, LYN, P3H4, PTK7, RPM3, S100A4, SYNM, SYNPO2, TAGLN, SERPIN1, SERPINB1, and VWA1 have been previously associated with neoplastic proteomes in humans [30,31,37–43]. Identification of 6 proteins that have previously been identified in human NENs (CKB, CLU, LGALS1, LYN, SERPINB1, TAGLN) serves as a proof-of-principle that the methodology was valid. Among these proteins, galectin-1 (LGALS1) may

**Fig 6. STRING network of differentially expressed downregulated proteins canine hepatobiliary NEN.** Log-ratios of protein abundance between cancer and normal tissues were mapped using a white-purple gradient. STRING network confidence cutoff is 0.7. Nodes connected by pink edges have interactions that have been experimentally validated. Image generated in Cytoscape StringApp.

presents a potential therapeutic target [31,44]. Anti-galectin pharmaceuticals have shown promise against cervical cancer, prostate cancer, lung and pancreatic neuroendocrine neoplasia in human medicine [30,31,45,46]. Pharmacokinetic studies are warranted to determine whether galectin-1 antagonists have appropriate safety profiles in canine patients.

Fifty percent of the differentially expressed upregulated proteins were localized to the exosomal cellular location. This is consistent with the transition to a neuroendocrine phenotype. Neuroendocrine cells use secretory vesicles containing neurotransmitters and biogenic amines for paracrine signaling [47]. However, these proteins were differentially expressed compared to a normal neuroendocrine organ, the adrenal gland. Therefore, this upregulation in expression may be represent a shift towards intercellular tumor signaling [48].

Enrichment analysis of the differentially expressed downregulated proteins clustered almost exclusively within metabolic pathways. The shift to glycolytic respiration in cancer is well characterized, and is supported by the apparent reduction in protein expression related to fatty acid beta oxidation in this data set. Malignant metabolic reprogramming has proven more complex that originally believed; therefore, these results prompt a more in-depth preclinical evaluation of the metabolic status of the hepatobiliary neuroendocrine tumor microenvironment to determine whether this feature can be therapeutically exploited [49].

One weakness of this work is its small scale. Collaborative efforts to achieve higher patient numbers are requisite. Yet, even this limited proteomic snapshot of 3 canine hepatobiliary neuroendocrine tumors highlights how a comparative approach using unbiased quantitative protein expression patterns can glean actionable data. The second major weakness of this work is that the tissue samples, normal and neoplastic, are an amalgam of cell types. An additional limitation of this work is the use of FFPE samples for proteome analysis. While fresh frozen samples are the gold standard for comparative proteomics, these are not routinely collected from veterinary patients. The primary flaws associated with formalin fixation include formaldehyde induced protein cross-linking and amino acid modifications, which leads to difficulties with protein profiling. However, recent improvements in mass spectrometry workflows have enabled FFPE proteome maps comparable to those derived from fresh frozen tissue. This study demonstrates that archived FFPE canine tumor samples can be a rich resource for biomarker discovery. Future work will focus on identifying the small, poorly characterized pool of normal neuroendocrine cells within the hepatobiliary niche and overcoming the technical challenges of single cell molecular characterization. Investigation of neoplastic proteome expression is not novel, yet cross-species comparisons of rare cancers may provide valuable insight into neoplastic homeostasis and hone molecular drug targets [50,51].

## Conclusions

In summary, using a comparative proteomic strategy, we have identified 32 upregulated and 126 downregulated differentially expressed proteins in canine hepatobiliary NENs. These data demonstrate that characteristic tumor protein signatures can be determined from the abundant archive of FFPE tissues and that promising biomarker candidates can be derived from a small number of samples.

## Supporting information

**S1 Fig. Transgelin immunofluorescent staining of canine gallbladder neuroendocrine neoplasm.** Nuclei are labeled by Hoescht stain (blue, left panel). Gall bladder neuroendocrine neoplasm demonstrates cytoplasmic positive transgelin (green, middle panel and right panel) immunofluorescence compared to negative control (normal canine liver), wherein only

endothelium expresses transgelin. Scale bars: 200 micrometers.
(TIF)

**S2 Fig. Galectin-1 immunofluorescent staining.** Nuclei are labeled by Hoescht stain (blue, left panels). Gall bladder neuroendocrine neoplasm demonstrates positive galectin-1 (red, middle and right top panels) immunofluorescence compared to negative control (normal canine kidney). Scale bars: 10 micrometers.
(TIF)

## Acknowledgments

We would like to thank Dr. Nadia Richmond for completing the immunofluorescent staining, Drs. Fabio Del Piero and Mariano Carossino for their histologic assessment of select cases, and Drs. Margaux Marclay and Ashley Hegler for medically managing the clinical cases.

## Author Contributions

**Conceptualization:** Joshua Sparago, Rudy W. Bauer, Daniel Paulsen, Bonnie Boudreaux, Andrea N. Johnston.

**Data curation:** Emi Sasaki, Mayzie Miller, Joshua Sparago, Rudy W. Bauer, Daniel Paulsen, Bonnie Boudreaux, Stephanie D. Byrum, Andrea N. Johnston.

**Formal analysis:** Tifini L. Batts, Chin-Chi Liu, Stephanie D. Byrum, Andrea N. Johnston.

**Funding acquisition:** Stephanie D. Byrum, Andrea N. Johnston.

**Methodology:** Stephanie D. Byrum, Andrea N. Johnston.

**Software:** Stephanie D. Byrum.

**Supervision:** Andrea N. Johnston.

**Validation:** Emi Sasaki, Chin-Chi Liu.

**Visualization:** Tifini L. Batts.

**Writing – original draft:** Tifini L. Batts, Andrea N. Johnston.

**Writing – review & editing:** Tifini L. Batts, Emi Sasaki, Chin-Chi Liu, Stephanie D. Byrum, Andrea N. Johnston.

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
