## [Decision Letter · Decision Letter 0]

28 Nov 2022

PONE-D-22-02231Neoplastic signatures: comparative proteomics of canine hepatobiliary neuroendocrine tumors to normal niche tissuePLOS ONE

Dear Dr. Batts,

Thank you for submitting your manuscript to PLOS ONE. After careful consideration, we feel that it has merit but does not fully meet PLOS ONE’s publication criteria as it currently stands. Therefore, we invite you to submit a revised version of the manuscript that addresses the points raised during the review process.

Both reviewers have raised some minor issues which should be considered by the authors.

We look forward to receiving your revised manuscript.

Kind regards,

Salvatore V Pizzo

Academic Editor

PLOS ONE

Journal Requirements:

” This work was supported in part by NIH grant 1P20GM135000-01A1 and the Louisiana State University, Department of Veterinary Clinical Sciences CORP grant. Proteomic analysis was performed by the IDeA National Resource for Quantitative Proteomics, funded by the National Institute of General Medical Sciences (NIGMS) grant R24GM137786. This study was also supported by the Arkansas Children's Research Institute, the Arkansas Biosciences Institute, and the Center for Translational Pediatric Research funded under the National Institutes of Health National Institute of General Medical Sciences (NIH/NIGMS) grant P20GM121293 and the National Science Foundation Award No. OIA-1946391.”

‘AJ - NIH grant 1P20GM135000-01A1 and the Louisiana State University, Department of Veterinary Clinical Sciences CORP grant

SB - Proteomic analysis was funded by the National Institute of General Medical Sciences (NIGMS) grant R24GM137786.

SB- Arkansas Children's Research Institute, the Arkansas Biosciences Institute, and the Center for Translational Pediatric Research funded under the National Institutes of Health National Institute of General Medical Sciences (NIH/NIGMS) grant P20GM121293 and the National Science Foundation Award No. OIA-1946391.”

Reviewers' comments:

Reviewer's Responses to Questions

**Comments to the Author**

1. Is the manuscript technically sound, and do the data support the conclusions?

Reviewer #1: Yes

Reviewer #2: Yes

2. Has the statistical analysis been performed appropriately and rigorously? 

Reviewer #1: Yes

Reviewer #2: I Don't Know

3. Have the authors made all data underlying the findings in their manuscript fully available?

Reviewer #1: Yes

Reviewer #2: Yes

4. Is the manuscript presented in an intelligible fashion and written in standard English?

Reviewer #1: Yes

Reviewer #2: Yes

5. Review Comments to the Author

Reviewer #1: The author compared the proteomes of 3 canine hepatobiliary neoplasms to normal hepatobiliary tissue, which is an attractive topic, as hepatobiliary neuroendocrine neoplasm is rare cancers in humans and primary hepatobiliary neoplasm omics studies is still lacking. The authors extracted proteins from canine hepatobiliary neoplasms and performed mass spectrometry analysis. The data showed that 32 proteins were up-regulated and 126 proteins were down-regulated. Among them, 6 up-regulated proteins were associated with non-hepatobiliary neuroendocrine tumors, 16 proteins were associated with neuroendocrine signal transduction, and 26 down-regulated proteins were enriched in altered metabolic pathways of cancer. However, I do not think at the current stage the article has a potential for consideration in the journal for the publication.

1. For screening up-regulated and down-regulated proteins in canine hepatobiliary neoplasms, in order to increase the persuasiveness of the article, it is recommended to add some experiments to verify the expression of significantly increased proteins.

2. I suggest, the clinical information should be added, such as patient classification.

3. Please add scale in Figure 2.

Reviewer #2: There is limited research on proteomics available in veterinary oncology. Furthermore, there are currently no studies evaluating proteomics in canine hepatobiliary neuroendocrine tumors. Thus, this manuscript begins to help fill the gaps in our understanding of these tumors. The introduction, materials and methods, results, and figures are complete and thorough.

I have one proposed revision for the discussion section. The limitations of utilizing FFPE samples as opposed to fresh frozen tissue samples should be addressed, as fresh frozen tissue is considered gold standard for comparative proteomics. Since FFPE samples are more common and overall reliable, I feel this information is still an important contribution to the current body of literature.

6. PLOS authors have the option to publish the peer review history of their article (what does this mean?). If published, this will include your full peer review and any attached files.

Reviewer #1: No

Reviewer #2: No

---

## [Author Response · Author response to Decision Letter 0]

2 Jan 2023

PONE-D-22-02231-R1

The authors appreciate the recommendations for improvement provided by the reviewers. Our responses are listed below each comment.

Journal Requirements:

Response: The templates have been reviewed. The manuscript and files have been modified to meet PLOS ONE’s style requirements.

” This work was supported in part by NIH grant 1P20GM135000-01A1 and the Louisiana State University, Department of Veterinary Clinical Sciences CORP grant. Proteomic analysis was performed by the IDeA National Resource for Quantitative Proteomics, funded by the National Institute of General Medical Sciences (NIGMS) grant R24GM137786. This study was also supported by the Arkansas Children's Research Institute, the Arkansas Biosciences Institute, and the Center for Translational Pediatric Research funded under the National Institutes of Health National Institute of General Medical Sciences (NIH/NIGMS) grant P20GM121293 and the National Science Foundation Award No. OIA-1946391.”

‘AJ - NIH grant 1P20GM135000-01A1 and the Louisiana State University, Department of Veterinary Clinical Sciences CORP grant

SB - Proteomic analysis was funded by the National Institute of General Medical Sciences (NIGMS) grant R24GM137786.

SB- Arkansas Children's Research Institute, the Arkansas Biosciences Institute, and the Center for Translational Pediatric Research funded under the National Institutes of Health National Institute of General Medical Sciences (NIH/NIGMS) grant P20GM121293 and the National Science Foundation Award No. OIA-1946391.”

Response: The funding related statements have been removed from the Acknowledgments section and the cover letter includes the funding information previously omitted from the funding statement. Thank you for changing the online submission on our behalf.

Response: The reference list has been reviewed. It is complete and correct. No cited papers have been retracted. 

Comments to the Reviewers

5.Review Comments to the Author

Reviewer #1: The author compared the proteomes of 3 canine hepatobiliary neoplasms to normal hepatobiliary tissue, which is an attractive topic, as hepatobiliary neuroendocrine neoplasm is rare cancers in humans and primary hepatobiliary neoplasm omics studies is still lacking. The authors extracted proteins from canine hepatobiliary neoplasms and performed mass spectrometry analysis. The data showed that 32 proteins were up-regulated and 126 proteins were down-regulated. Among them, 6 up-regulated proteins were associated with non-hepatobiliary neuroendocrine tumors, 16 proteins were associated with neuroendocrine signal transduction, and 26 down-regulated proteins were enriched in altered metabolic pathways of cancer. However, I do not think at the current stage the article has a potential for consideration in the journal for the publication.

1. For screening up-regulated and down-regulated proteins in canine hepatobiliary neoplasms, in order to increase the persuasiveness of the article, it is recommended to add some experiments to verify the expression of significantly increased proteins.

Response: Thank you for this recommendation to improve the persuasiveness of the manuscript. Upregulation of two proteins of interest, transgelin and galectin-1, were validated by immunofluorescence microscopy (S1 and S2 Figs). Transgelin and galectin-1 have been previously associated with human neuroendocrine tumors and thus provide proof of principle. 

2. I suggest, the clinical information should be added, such as patient classification.

Response: In addition to patient signalment, which was originally included, the following clinical information has been added to the text (lines 114-120): “The Boston terrier was asymptomatic with a normal complete blood count and serum biochemistry. The Doberman pinscher presented for hematemesis after dietary indiscretion and had a mild elevation in GGT and cholesterol with moderate to marked elevations in ALP and ALT, respectively. No extrahepatobiliary masses were identified on abdominal ultrasound or during surgery. The patient with the liver NEN was a 12-year-old, male neutered Rat terrier with a history of lethargy, polyuria, polydipsia, and diarrhea. He had a normal complete blood count and moderate elevations of ALP and ALT on serum biochemistry.”

3. Please add scale in Figure 2.

Response: A scale bar has been added to Figure 2.

Reviewer #2: There is limited research on proteomics available in veterinary oncology. Furthermore, there are currently no studies evaluating proteomics in canine hepatobiliary neuroendocrine tumors. Thus, this manuscript begins to help fill the gaps in our understanding of these tumors. The introduction, materials and methods, results, and figures are complete and thorough.

I have one proposed revision for the discussion section. The limitations of utilizing FFPE samples as opposed to fresh frozen tissue samples should be addressed, as fresh frozen tissue is considered gold standard for comparative proteomics. Since FFPE samples are more common and overall reliable, I feel this information is still an important contribution to the current body of literature.

Reviewer: Thank you for this suggestion. A discussion of the limitations of using FFPE tissue compared to fresh tissue has been included (lines 359-367): “An additional limitation of this work is the use of FFPE samples for proteome analysis. While fresh frozen samples are the gold standard for comparative proteomics, these are not routinely collected from veterinary patients. The primary flaws associated with formalin fixation include formaldehyde induced protein cross-linking and amino acid modifications, which leads to difficulties with protein profiling. However, recent improvements in mass spectrometry workflows have enabled FFPE proteome maps comparable to those derived from fresh frozen tissue. This study demonstrates that archived FFPE canine tumor samples can be a rich resource for biomarker discovery.”

---

## [Decision Letter · Decision Letter 1]

12 Jan 2023

Neoplastic signatures: comparative proteomics of canine hepatobiliary neuroendocrine tumors to normal niche tissue

PONE-D-22-02231R1

Dear Dr. Batts,

We’re pleased to inform you that your manuscript has been judged scientifically suitable for publication and will be formally accepted for publication once it meets all outstanding technical requirements.

Kind regards,

Salvatore V Pizzo

Academic Editor

PLOS ONE

Additional Editor Comments (optional):

Reviewers' comments:

Reviewer's Responses to Questions

**Comments to the Author**

1. If the authors have adequately addressed your comments raised in a previous round of review and you feel that this manuscript is now acceptable for publication, you may indicate that here to bypass the “Comments to the Author” section, enter your conflict of interest statement in the “Confidential to Editor” section, and submit your "Accept" recommendation.

Reviewer #1: All comments have been addressed

2. Is the manuscript technically sound, and do the data support the conclusions?

Reviewer #1: Yes

3. Has the statistical analysis been performed appropriately and rigorously? 

Reviewer #1: I Don't Know

4. Have the authors made all data underlying the findings in their manuscript fully available?

Reviewer #1: Yes

5. Is the manuscript presented in an intelligible fashion and written in standard English?

Reviewer #1: (No Response)

6. Review Comments to the Author

Reviewer #1: (No Response)

7. PLOS authors have the option to publish the peer review history of their article (what does this mean?). If published, this will include your full peer review and any attached files.

Reviewer #1: No

---

## [Editor Report · Acceptance letter]

16 Jan 2023

PONE-D-22-02231R1 

Neoplastic signatures: comparative proteomics of canine hepatobiliary neuroendocrine tumors to normal niche tissue 

Dear Dr. Batts:

I'm pleased to inform you that your manuscript has been deemed suitable for publication in PLOS ONE. Congratulations! Your manuscript is now with our production department. 

Kind regards, 

on behalf of

Dr. Salvatore V Pizzo 

Academic Editor

PLOS ONE